# *Cinnamomum verum* Bark Extract Mediated Green Synthesis of ZnO Nanoparticles and Their Antibacterial Potentiality

**DOI:** 10.3390/biom10020336

**Published:** 2020-02-19

**Authors:** Mohammad Azam Ansari, Mahadevamurthy Murali, Daruka Prasad, Mohammad A. Alzohairy, Ahmad Almatroudi, Mohammad N. Alomary, Arakere Chunchegowda Udayashankar, Sudarshana Brijesh Singh, Sarah Mousa Maadi Asiri, Bagepalli Shivaram Ashwini, Hittanahallikoppal Gajendramurthy Gowtham, Nataraj Kalegowda, Kestur Nagaraj Amruthesh, Thimappa Ramachandrappa Lakshmeesha, Siddapura Ramachandrappa Niranjana

**Affiliations:** 1Department of Epidemic Disease Research, Institutes for Research and Medical Consultations (IRMC), Imam Abdulrahman Bin Faisal University, Dammam 31441, Saudi Arabia; 2Applied Plant Pathology Laboratory, Department of Studies in Botany, University of Mysore, Manasagangotri, Mysore 570006, Karnataka, India; botany.murali@gmail.com (M.M.); gajendramurthygowtham@gmail.com (H.G.G.); knataraj922@gmail.com (N.K.); dr.knamruthesh@botany.uni-mysore.ac.in (K.N.A.); 3Department of Physics, B.M.S. Institute of Technology, Bangalore 560 064, India; darukap@bmsit.in; 4Department of Medical Laboratories, College of Applied Medical Sciences, Qassim University, Qassim 51431, Saudi Arabia; dr.alzohairy@gmail.com (M.A.A.); aamtrody@qu.edu.sa (A.A.); 5National Center for Biotechnology, Life Science and Environmental Research Institute, King Abdulaziz City for Science and Technology, P.O. Box 6086, Riyadh, Saudi Arabia; malomary@kacst.edu.sa; 6Department of Studies in Biotechnology, Manasagangotri, University of Mysore, Mysuru- 570 006, Karnataka, India; ac.uday@gmail.com (A.C.U.); brijeshrajput.bt@gmail.com (S.B.S.); 7Department of Biophysics, Institutes for Research and Medical Consultations (IRMC), Imam Abdulrahman Bin Faisal University, Dammam 31441, Saudi Arabia; smasiri@iau.edu.sa; 8Department of Microbiology, Sri Siddhartha Medical College, Tumkuru – 572107, Karnataka, India; ashwinibs17@gmail.com; 9Department of Microbiology and Biotechnology, Jnana Bharathi Campus, Bangalore University, Bangalore 560056, India

**Keywords:** green synthesis, ZnO-NPs, *Cinnamomum verum*, GC-MS, antibacterial activity

## Abstract

*Cinnamomum verum* plant extract mediated propellant chemistry route was used for the green synthesis of zinc oxide nanoparticles. Prepared samples were confirmed for their nano regime using advanced characterization techniques such as powder X-ray diffraction and microscopic techniques such as scanning electron microscopy and transmission electron microscopy. The energy band gap of the green synthesized zinc oxide (ZnO)-nanoparticles (NPs) were found between 3.25–3.28 eV. Fourier transmission infrared spectroscopy shows the presence of Zn-O bond within the wave number of 500 cm^−1^. SEM images show the specific agglomeration of particles which was also confirmed by TEM studies. The green synthesized ZnO-NPs inhibited the growth of *Escherichia coli* and *Staphylococcus aureus* with a minimum inhibitory concentration (MIC) of 125 µg mL^−1^ and 62.5 µg mL^−1^, respectively. The results indicate the prepared ZnO-NPs can be used as a potential antimicrobial agent against harmful pathogens.

## 1. Introduction

Most of the antibacterial compounds are of bulk complex materials having high toxicological side effects [1,2,3,4,5]. Recent studies on antibacterial activity using nanoparticles (NPs) have shown to reduce the side effects, although the data are both limited and controversial [6,7,8,9]. The antibacterial activity of NPs is attributed to their nano scale dimensions, changed electronic/optical transitions and high surface to volume ratio which leads to improved surface chemistry and quantum confinement effects [10,11]. Nano research has created various possibilities in surface-modified applications [12,13,14]. Among the metal oxide nanoparticles, zinc oxide (ZnO) NPs are one of the most extensively studied nanoparticles due to its wide band gap energy (~3.37 eV) making them applicable in the field of photonics and nanoelectronics fields including semiconductor production. These ZnO nanoparticles also act as an agent to block UV radiation making them useful in the field of by dermatitis for the prevention of skin diseases [15,16]. ZnO nanoparticles can be obtained through various methods, viz., thermal evaporation, magnetic sputtering, solution combustion, molecular beam epitaxial, pulsed laser deposition, etc. [17,18,19,20]. Nowadays it has been proved that the green synthesis of NPs is eco-friendly, low cost and is less toxic when compared to other conventional synthesis methods [21,22]. Apart from the above-mentioned properties, these ZnO-NPs synthesized through plant extracts are also known to possess efficient antibacterial properties [1,3,10,21].

*Cinnamomum verum* J. Presl is an evergreen tree belonging to the family *Lauraceae*, abundantly found in Sri Lanka and other parts of Asian countries [23,24]. The different parts of the plant are used in Indian traditional medicine for the treatment of antibacterial, antifungal and anti-inflammatory, etc. [25,26,27,28].The cinnamon bark extract is one of the major bioactive compounds which have been reported to possess many biological activities such as antibacterial, antibiofilm, anthelmintic, anticancer and antifungal activity [29,30,31,32,33].Due to its vast biological potentialities, in the present study, *C. verum* bark extract was used for the green synthesis of ZnO NPs. Structural, morphological and optical studies were investigated by the advanced characterization techniques. Antibacterial activity was studied by the disc diffusion and broth microdilution method and supported the observations by the fluorescence microscopic method.

## 2. Materials and Synthesis Method

### 2.1. Collection of Plant Material and Green Synthesis of ZnO-NPs

*Cinnamomum verum* plant was collected from Gandhi Krishi Vigyan Kendra (GKVK), University of Agricultural Sciences, Bangalore, India (12°35′ N latitude and 77°35′ E longitude and at an elevation of 930 m above mean sea level) based on an ethno botanical survey. The collected plant material was authenticated by a Prof. M.S. Sudarshana, Taxonomist, Department of Studies in Botany, University of Mysore. *C. verum* bark material (50 g) was rinsed in de-ionized water (100 mL), dried at 35–40 °C for 48 h using an oven and further powdered by a sterile electric blender. The powdered *C. verum* bark was extracted with hexane (1:5, *w*/*v*) and the resultant extract was filtered through sterile Whatman No. 1 filter paper. The obtained filtrate was used for the green synthesis of ZnO-NPs. To predict the major compounds of *C. verum* bark extract, gas chromatography-mass spectrophotometer (GC-MS) analyzed was carried out using Shimadzu GC-MSQP2010 (Tokyo, Japan) equipped with a mass selective detector. Green synthesis of ZnO-NPs was carried out by the addition of zinc nitrate hexahydrate (2 g) to *C. verum* hexane extract at different concentrations (5, 10, 15, 20 and 25 mL) and kept on a magnetic stirrer (45–60 °C) until the solution becomes a paste. The resultant mixture was transferred to a silica crucible (individually) and kept in a muffle furnace for 2 h at 400 ± 10 °C. The resultant, ZnO nanoparticles were named ZnO(a), ZnO(b), ZnO(c), ZnO(d) and ZnO(e), respectively for the ratio of *C. verum* to ZnO-NPs.

### 2.2. Characterization of ZnO NPs

UV-Vis spectral analysis of a sonicated sample of ZnO-NPs (1 mg mL^−1^ in de-ionized water) was performed using the Systronic model (Double beam spectrometer 2203) at room temperature in the range of 200 to 800 nm. The green synthesized ZnO-NPs were characterized by powder X-ray Diffractometer (PXRD, Philips PW 3050/10 model, Cuk_α_ (1.541Ǻ) radiation. The data were collected over a 2θ range from 25° to 80° with a scanning rate of 0.02° with a counting time of 1 s per step. The topographic features of each ZnO NPs were visualized with an SEM (Carl Zeiss, Jena, Germany) in high vacuum mode at 15 kV. To determine the size of the ZnO NPs transmission electron microscopy (TEM) examination was performed on Philips model CM 200 instrument. The characteristic functional groups present in the ZnO NPs were carried out in FT-IR spectroscopy (Perkin Elmer Spectrum 1000) in attenuated total reflection mode (spectral range of 400 to 4000 cm^−1^) [34].

### 2.3. Antibacterial Activity

The bacterial strains of *Staphylococcus aureus* (MTCC 7443) (gram-positive) and *Escherichia coli* (MTCC 7410) (gram-negative) cultures were procured from Microbial Type Culture Collection (IMTECH, Chandigarh, India) and sub-cultured and maintained on Muller Hinton medium. The stock solution of ZnO-NPs (80 mg mL^−1^) was subjected to sonication (15 min) before subjecting to antibacterial studies.

#### 2.3.1. Disc Diffusion Method

The disc diffusion method was used to test the antibacterial efficacy of green synthesized ZnO-NPs against the test bacterial pathogens [35]. About 100 µL (1.5 × 10^8^ CFU mL^−1^) of the active culture of each of the bacterium (grown on Muller Hinton broth medium for 12 h at 37 °C) was spread uniformly on Muller Hinton medium in Petri plates by a sterile glass spreader. The dry 6 mm sterile discs impregnated with ZnO-NPs (25 µL) (2 mg disc^−1^) was placed on Muller Hinton plates uniformly seeded with the test bacterial inoculum. The sterile discs impregnated with 25 µL streptomycin (25 µg disc^−1^) and sterile distilled water (SDW) served as a positive and negative control, respectively. The inoculated Petri plates were kept in an incubator for 24 h (at 37 °C) and the inhibition zone formed around the discs was measured and tabulated.

#### 2.3.2. Minimum Inhibitory Concentration (MIC)

The MIC of green synthesized ZnO-NPs was also determined by following the broth micro dilution technique [36]. The green synthesized ZnO-NPs (2 mg mL^−1^) in SDW served as a stock solution. About 100 μL of Muller Hinton broth medium was added into each well of the 96-well ELISA plate followed by an equal volume of ZnO-NPs stock solution and 2-fold serially diluted (2 to 0.003 mg mL^−1^). To each well, 10 μL of test bacterial (1.5 × 10^8^ CFU mL^−1^) suspension was added and the plates were incubated in an incubator for 24 h (at 37 °C). The incubated ELISA plates were measured for absorbance (at 620 nm) using ELISA plate reader (LabTech 4000). The MIC was also confirmed by adding (10 μL well-1) 2,3,5- triphenyl tetrazolium chloride (TTC) (2 mg mL^−1^) and incubating for 30 min under dark conditions. The change in color observed in the wells was considered as the MIC.

#### 2.3.3. Morphological Evaluation of Bacterial Cells by Fluorescence Microscopy

Fluorescence microscopy was used to distinguish the live and dead cells upon treatment with ZnO-NPs with minor modifications [37]. To 100 µL bacterial cell suspensions (1.5 × 10^8^ CFU mL^−1^), 50 µL of ZnO-NPs (2 mg) were added and mixed thoroughly and kept for incubation at 37 ± 2 °C for 24 h, while the untreated cells served as control. After incubation, the mixture was centrifuged (at 5000 rpm for 5 min at 4 °C) and the pellet was washed thrice with phosphate buffer saline (PBS). Each of the samples obtained was mixed with fluorescent dye solution of ethidium bromide (EB) and acridine orange (AO) at 1:1 ratio (AO/EB 100 μg mL^−1^) and incubated for 30 min. After incubation, the mixture was centrifuged and again rinsed with PBS, about 5 µL bacterial cell suspension was placed on sterilized grease free slide and covered by glass cover slip and viewed at 40× in Carl Zeiss fluorescence microscope (Lawrence and Mayo, Germany) with excitation filter 430 to 470 nm.

### 2.4. Statically Analysis

The antibacterial studies were carried out with four replicates for each experiment and were statistically analyzed by subjecting to arcsine transformation and analysis of variance (ANOVA) using SPSS, version 17 (SPSS Inc., Chicago, IL, USA). The significant differences between the treatments mean values were determined by honestly significant difference (HSD) obtained by Tukey’s test at *p* ≤ 0.05 levels.

## 3. Results and Discussion 

### 3.1. Collection of Plant Material and Green Synthesis of ZnO-NPs

The gas chromatography mass spectrometer (GC-MS) analytical method was used to detect the secondary metabolites present in the bark extract of *C. verum*. From the results of GC-MS analysis, eugenol was detected as one of the major components in the extract (Appendix A) which is reported to possess many biological activities [30,31,32,33]. It may be noted that, to date, the exact mechanism involved during the formation of nanoparticles from plant extract is not reported. Hence, in the present study, we have provided aplausible mechanism involved during the formation of ZnO-NPs by capping of eugenol present in the plant extract (Figure 1). It is noted from previous studies that zinc ions (Zn^2+^) will cap with available secondary metabolites of plant extract to form ZnO-NPs [38,39].

### 3.2. Characterization of ZnO NPs

The UV-Visible absorption spectra showed a distinct absorption peak around at ~370 to 375 nm (Figure 2 and Appendix A) with band gap energy between 3.25 to 3.28 eV (Table 1). It has been previously reported that the ZnO-NPs possess a distinct absorption spectrum between 280 to 400 nm, and due to this characteristic feature, these nanoparticles have been employed in pharmaceutical and biological applications [40,41].The PXRD profiles for the ZnO prepared with different concentrations of *C. verum* hexane extract are presented in Figure 3A. Rietveld structural refinement was performed by using the *Full-prof* program to analyze the crystalline structure and unit cell of the fabricated ZnO-NPs. The obtained Rietveld refinement and microstress results of the synthesized particles are summarized in Table 1 and the fitment profile for *C. verum* ZnO (c) is depicted in Figure 3B.The PXRD profile showed the diffraction peaks which are in alignment to the hexagonal Wurtzite phase of zinc oxide (JCPDS card No. 89–1397) which includes Miller indices (100), (002), (101), (102), (110), (103), (200), (112), (201), (004) and (202) for various concentration (ZnO(a), ZnO(b), ZnO(c), ZnO(d) and ZnO(e)). In addition, all the diffraction peaks were well assigned to the hexagonal wurtzite phase of ZnO (JCPDS card No. 36–1451). The presence of the sharp high intensity peaks indicates high crystallinity of the nanoparticles. The crystallite size and strain present in the ZnO were estimated by using both Debye–Scherrer’s (DS) formula and W–H approach and are listed in Table 1. Statistical validity provided with Rietveld refinement showed the goodness of fit (GoF) nearer to one indicates the better fitting and without any impurities.SEM images of all the green synthesized ZnO-NPs are shown in Figure 4A–E. All the SEM images showed the agglomeration of ZnO-NPs. Transmission electron microscopy (TEM) images of ZnO-NPs showed the nano-regime as showed in Figure 5A, while Figure 5B shows the selected area electron diffraction (SAED) pattern revealing the green synthesized ZnO-NPs are of polycrystalline in nature. All the PXRD peaks (Miller indices) match with the SAED pattern of the green synthesized nanoparticles. D-spacing and the periodic arrangement of ZnO-NPs are shown in Figure 6C and the estimated d-spacing value for the sample was found to be 0.32 nm thereby confirming the clear separation of the inter-planar spacing. To confirm the phase transformation and purity of the ZnO, FT-IR spectra were recorded as showed in Figure 6. All the FT-IR spectra showed a wide band at 3311 cm^−1^ thereby indicating the presence of surface hydroxyl groups (Appendix A). Distinct bands observed around 496 cm^−1^ relates to the stretching vibrations and the bending modes of Zn–O groups.

### 3.3. Antibacterial Activity

#### 3.3.1. Disc Diffusion Method

The green synthesized ZnO-NPs significant inhibition against the test pathogens (*S. aureus* and *E. coli*) compared to control. ZnO-NPs offered a maximum zone of inhibition of 16.75 mm and 13.25 mm was observed against *S. aureus* and *E. coli*, respectively. From the study, it was observed that the gram-positive (*S. aureus*) bacterium was more sensitive to the green synthesized ZnO-NPs compared to the gram-negative (*E. coli*) bacterium. The obtained results are in good agreement with the findings of other researchers, wherein it was observed that gram-positive (*S. aureus* and *Bacillus subtilis*) bacteria were more sensitive to ZnO-NPs compared to gram-negative (*E. coli*a and *Pseudomonas aeruginosa*) bacteria [42,43,44,45]. Further, it has been noted that Zn^2+^ ions will stimulate the growth of both *E. coli* and *S. aureus* but the concentration required by *S. aureus*is lesser (micro molar range) [46,47] compared *E. coli* (milli molar range) [48,49] and these differences in the metabolism-dependent processes of *E. coli* and *S. aureus* may have a direct role in the observed difference in toxicity thresholds of these organisms [49]. Further, it was also noted that the positive control (streptomycin 25 μg disc^−1^) offered a zone of inhibition of 24.25 and 22 mm against *S. aureus* and *E. coli*, respectively while all the test samples were unable to inhibit the growth of test pathogens (Figure 7 and Table 2).The results of the disc diffusion study showed that except for ZnO-NPs and streptomycin all other treatments were unable to inhibit the growth of *S. aureus* and *E. coli*. The results of the study are in line with the findings of Mahendra et al. [50], wherein biosynthesized ZnO-NPs offered better inhibition to test pathogens compared to plant extract. The antibacterial properties of the biosynthesized ZnO-NPs have been attributed to the disruption of the cell membrane of the pathogen possibly due to the generation of hydrogen peroxide (H_2_O_2_) as they can infiltrate into the cell membrane thereby causing severe damage to bacteria [51].

#### 3.3.2. Minimum Inhibitory Concentration (MIC)

Further, the MIC of ZnO-NPs was determined by the broth microdilution method along with control. From the results, it was noted that the green synthesized ZnO-NPs showed a MIC of 62.5 µg mL^−1^ and 125 µg mL^−1^, while the streptomycin showed and MIC of 7.8 µg mL^−1^ for *S. aureus* and *E. coli*, respectively (Table 2 and Appendix A). The results of the present study are in accordance with the findings of many other researchers wherein plant extract mediated synthesis of ZnO-NPs offered significant MIC activity below 0.5 mg mL^−1^ against both *E. coli* and *S. aureus* [50,52].

#### 3.3.3. Morphological Evaluation of Bacterial Cells by Fluorescence Microscopy

The morphological changes that occurred during the inhibition of bacterial pathogens were carried out by selective staining for live and dead cell analysis. Among the fluorescent nucleic acid dyes used, AO is known to stain both live and dead cells, while EB can penetrate only cells with damaged cell membranes [53]. The fluorescent dye analysis confirmed that the morphology of the *E. coli*and *S. aureus*cells wasdamaged upon interaction with the green synthesized ZnO-NPswhich was confirmed by the presence of reddish-orange cells which cell membrane damage (Figure 8b,d). In addition, the appearance of green fluoresced *E. coli* and *S. aureus* cells which were treated with SDW confirmed the intact cell wall structures (Figure 8a,c). Similar to the study, Mahendra et al. [50] and Hameed et al. [54] have also used AO/EB dye to study the live and dead cells of pathogenic bacteria upon interaction with biosynthesized ZnO-NPs from plant extract. In addition, Table 3 shows the importance of the ZnO-NPs and their inhibitory effects against the harmful pathogens including *E. coli* and *S. aureus.* The table represents the method of preparation, crystallite size, morphology, etc., of ZnO-NPs and the results of the present study are also compared.

## 4. Conclusions

In conclusion, the green synthesized ZnO-NPs from the hexane extract *C. verum* bark were evaluated for its antibacterial efficacy against *S. aureus* and *E. coli*.PXRD and TEM results confirm the crystallites size as ~45 nm. The energy bandgap of the compounds is in the range of 3.25–3.28 eV confirms the wide bandgap nature of the samples and the results obtained are in proper alignment with the reported literature. Rietveld refinement technique supported the hexagonal Wurtzite phase of ZnO with the goodness of fitment very close to one. There is a small shift in the major PXRD peak indicates the orientation of (101) plane due to plant extract. Minimum inhibitory concentration of ZnO(c) NPs was 62.5 µg mL^−1^ and 125 µg mL^−1^ for *S. aureus* and *E. coli*, respectively.

## Figures and Tables

**Figure 1 biomolecules-10-00336-f001:**
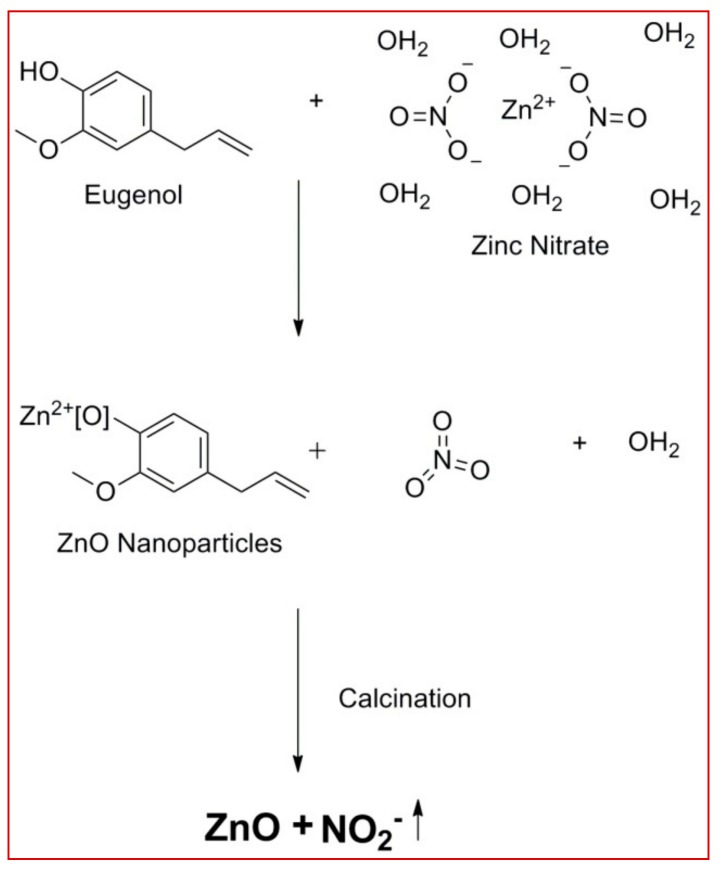
The plausible mechanism involved during the formation of zinc oxide (ZnO)-nanoparticles (NPs) from the bark extract of *C. verum.*

**Figure 2 biomolecules-10-00336-f002:**
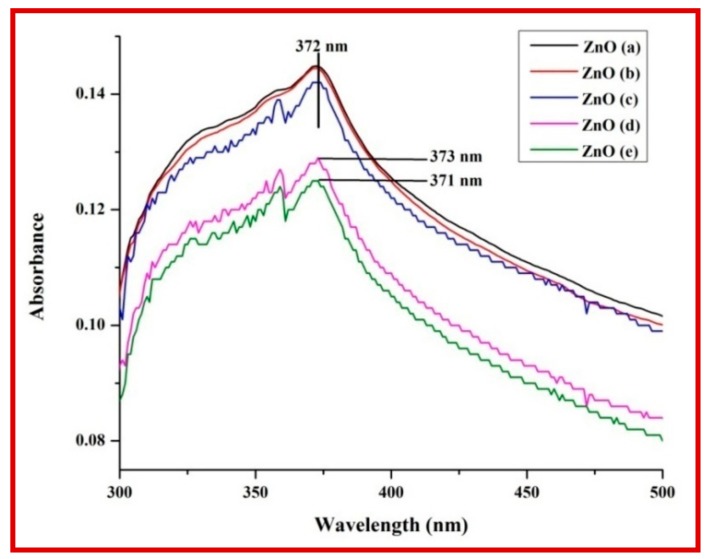
UV-visible spectra of various ZnO samples prepared using different concentrations of *C. verum.*

**Figure 3 biomolecules-10-00336-f003:**
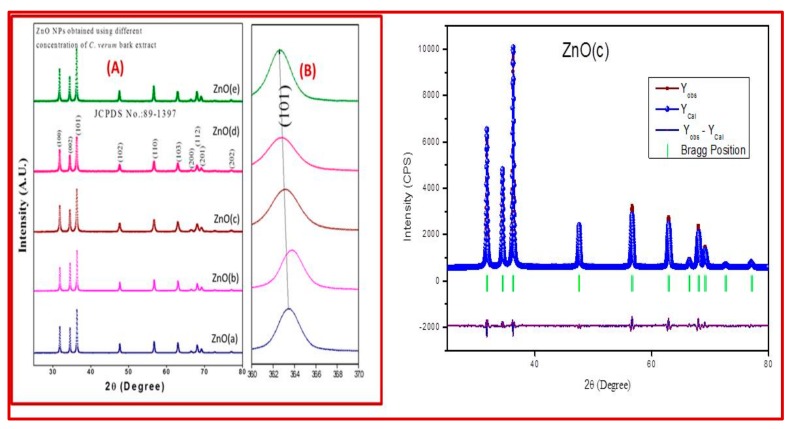
PXRD patterns of ZnO-NPs (**A**) and shift in the (101) plane obtained using various concentrationsof *C. verum* plant extract (**B**); Rietveld refined fitment showed for the ZnO(c) (**C**).

**Figure 4 biomolecules-10-00336-f004:**
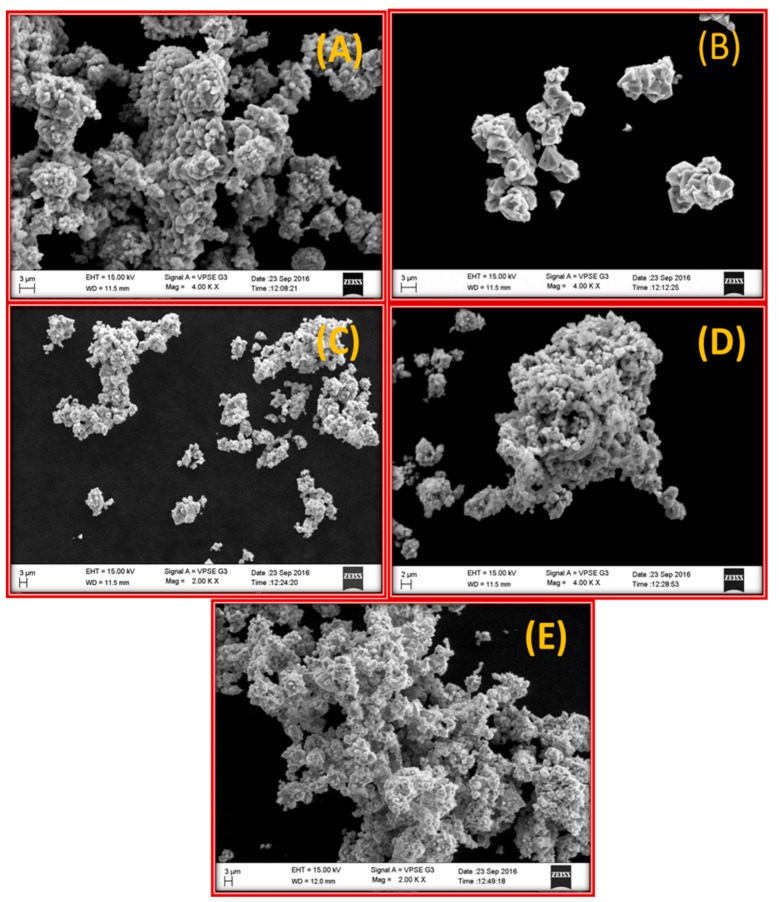
SEM micrograph ofZnO-NPs obtained using various concentration of *C. verum*plant extract (**A**) ZnO(a); (**B**) ZnO(b); (**C**) ZnO(c); (**D**) ZnO(d); (**E**) ZnO(e).

**Figure 5 biomolecules-10-00336-f005:**
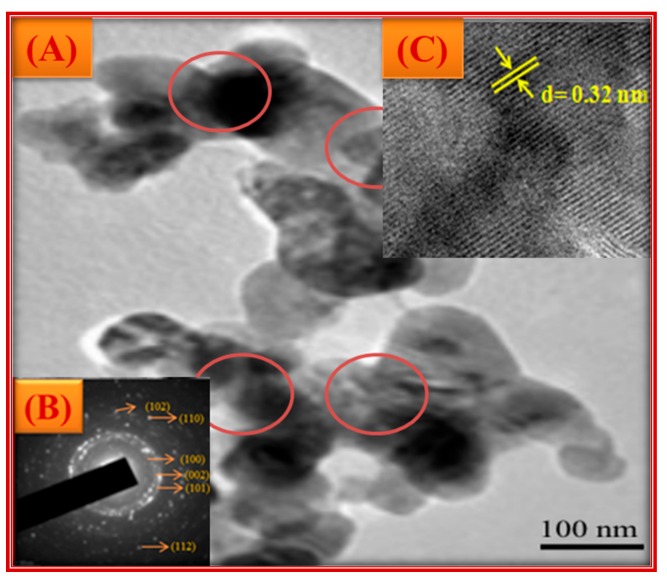
TEM image of ZnO(c) (**A**); SAED pattern of ZnO NPs (**B**); High-resolution transmission electron microscopy (HRTEM) with d-spacing (**C**).

**Figure 6 biomolecules-10-00336-f006:**
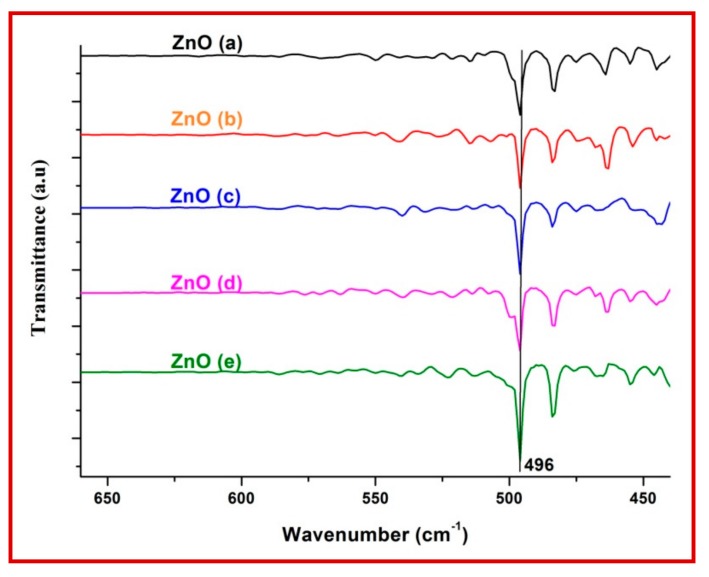
FT-IR Spectra of ZnO-NPs prepared using different concentrations of *C. verum* extract.

**Figure 7 biomolecules-10-00336-f007:**
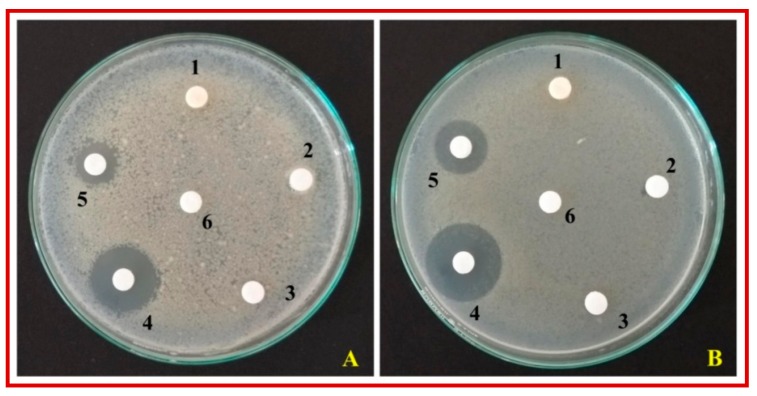
Antibacterial activity of ZnO-NPs green synthesized using *C. verum* hexane extract. (**A**): *E. coli*; (**B**): *S. aureus*; 1: *C. verum* hexane extract; 2: Hexane; 3: Hexane + zinc nitrate hexahydrate; 4: Streptomycin; 5: ZnO-NPs; 6: Sterile distilled water.

**Figure 8 biomolecules-10-00336-f008:**
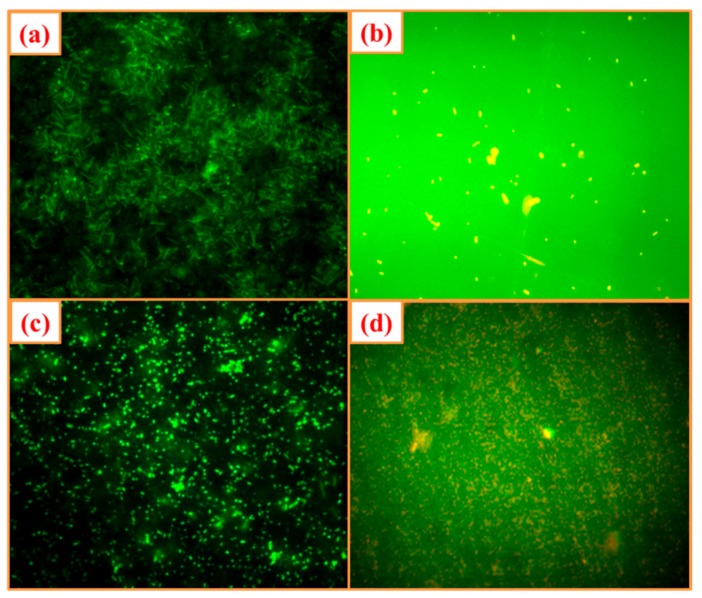
Fluorescent microscopic (40×) images of (**a**) *E. coli* and (**c**) *S. aureus* untreated control bacterial cells and (**b**) *E. coli* and (**d**) *S. aureus*treated with ZnO-NPs. In both series, green dots represent live bacterial cells and yellow/orange dots represent dead cells.

**Table 1 biomolecules-10-00336-t001:** Lattice parameters, crystallite sizes, micro stress and energy band gap values of ZnO prepared by various *C. verum* concentrations.

Compound	Lattice Parameters			Crystallite Size in nm		Micro Stress 𝞮_hkl_(10^3^)Nm^−2^	Energy Bandgap (E_g_) in eV
	a (Å)	b (Å)	V(Å)^3^	D-S Approach	Williamson-Hall Approach		
ZnO(a)	2.649	3.666	33.714	40	42	29.77	3.25
ZnO(b)	3.104	4.073	34.232	36	39	25.06	3.27
ZnO(c)	3.105	4.756	41.151	33	36	29.02	3.27
ZnO(d)	3.103	4.072	34.265	30	31	30.97	3.28
ZnO(e)	3.098	4.063	34.011	42	48	39.77	3.28

**Table 2 biomolecules-10-00336-t002:** Antibacterial activity of ZnO-NPs green synthesized using *C. verum* hexane extract.

Test Samples	Zone if Inhibition (mm)		MIC (µg mL^−1^)	
	*S. aureus*	*E. coli*	*S. aureus*	*E. coli*
Plant Extract	NA	NA	NA	NA
Hexane	NA	NA	NA	NA
ZnO-NPs(c)	16.75 ± 0.47	13.25 ± 0.75	62.5	125
Hexane + Zn(NO_3_)_2_ 6H_2_O	NA	NA	NA	NA
Streptomycin	24.25 ± 0.62	22.00 ± 0.40	15.62	15.62
Sterile Distilled Water	NA	NA	NA	NA

Values are means of four independent replicates (n = 4) and ± indicate standard errors.

**Table 3 biomolecules-10-00336-t003:** Comparison of obtained antibacterial results of prepared samples with the literature.

Nanoparticles	Bacteria	Zone of Inhibition/MIC/MBC	Plant	Method of Synthesis	Crystal Size	Morphology	Reference
ZnO NPs	*Escherichia coli*	4.2 mm	Arabic gum	Sol–gel method	16 nm	Spherical	[55]
ZnO NPs	*B. subtilis* *S. epidermidis* *S. aureus* *E. coli* *S. marcescens*	8 mm 16 mm 9 mm 9 mm 8.5 mm	*Pichia kudriavzevii*	Extracellular Synthesis	~10–61 nm	hexagonal wurtzite	[56]
ZnO quantum dots	*E. coli*	15.69 mm	*Eclipta alba*	Solution combustion method	~6 nm	Spherical	[57]
ZnO NPs	*S. epidermidis* *E. aerogenes*	14 mm 9 mm	*Amaranthus caudatus*	Solution combustion method	-	Spherical	[58]
ZnO NPs	*S. aureus* *E. coli* *P. aeruginosa*	31 mm 31 mm 23 mm	*Trifoliumpratense*	Solution combustion method	190 nm	Agglomerated	[59]
ZnO NPs	*S. thalpophilum* *Ochrobactrum* sp. *Achromobacter* sp. *Sphingobacterium*sp. *Acinetobacter* sp. *Ochrobactrum* sp.	1.5 mm 2.2 mm 1.2 mm 3 mm 1.7 mm 4 mm	*Boswelliaovalifoliolata*	Filtration method	20.3 nm	Spherical	[60]
ZnO NPs	*M. tuberculosis*	12.5 µg mL^−1^ (MIC)	*Limoniaacidissima*	Solution combustion method	12–53 nm	Spherical	[61]
ZnO NPs	*E. coli* *S. aureus* *P. aeruginosa*	2200 µg mL^−1^ (MIC) 2400 µg mL^−1^ (MBC)	*Aloe vera*	Solution combustion method	8–18 nm	Spherical, oval and hexagonal	[62]
ZnO NPs	*S. aureus* *S. paratyphi* *V. cholerae* *E. coli*	18 mm 6 mm 11 mm 7 mm	*Solanum nigrum*	Solution combustion method	29.79 nm	Quasi-spherical	[25]
ZnO NPs	*K. aerogenes* *E. coli* *P. desmolyticum* *S. aureus*	9.67 mm 4.67 mm 4 mm 4.67 mm	*Cassia fistula*	Solution combustion method	5–15 nm	Hexagonal Wurtzite structure	[63]
ZnO NPs	*S. aureus,* *S. pyogenes* *E. coli*	100 µg mL^−1^ (MIC)	*Azadirachtaindica*	Filtration	9.6–25.5 nm	Spherical	[64]
ZnO NPs	*S. aureus* *E. coli*	29 mm 25 mm	*Pongamiapinnata*	Solution combustion method	100 nm	Spherical	[65]
ZnO NPs	*S. aureus* *C. ablicans*	10 mm 10 mm	*Zingiberofficinale*	Solution combustion method	23–25 nm	Spherical	[66]
ZnO NPs	*S. aureus* *E. coli*	62.5 µg mL^−1^ (MIC) and 125 µg mL^−1^ (MIC)	*C. verum*	Green method	~45 nm	Hexagonal Wurtzite	Present paper
ZnO NPs	*C. albicans*	250 µg mL^−1^ (MIC)	*Crinum latifolium*	Green method	10–30 nm	hexagonal and spherical	[67]
ZnO-NPs	*S. aureus, B. subtilis, E. coli and S. typhi*	19–22 mm	*C. candalebrum*	Green method	12–35 nm	Hexagonal Wurtzite	[68]

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
