# Peer review of "Cinnamomum verum Bark Extract Mediated Green Synthesis of ZnO Nanoparticles and Their Antibacterial Potentiality"

_biomolecules, 2020, doi:10.3390/biom10020336_

Round 1

Reviewer 1 Report

The manuscript entitled „Cinnamomum verum bark extract mediated green synthesis of ZnO nanoparticles and characterization of their enhanced antibacterial activity” authored by Mohammad Azam Ansari, Thimappa Ramachndrappa Lakshmeesha, Murali Mahadevamurthy, Daruka Prasad, Mohammad A. Alzohairy, Mohammad N. Alomary, Udayashankar Arakere Chunchegowda, Sarah Mousa Maadi Asiri, Sudarshana Brijesh Singh, Bagepalli Shivaram Ashwini, Hittanahallikoppal Gajendramurthy Gowtham, Amruthesh Kestur Nagaraj and Siddapura Ramachandrappa Niranjana is an improved version of the text reviewed by me few months ago. The text is greatly improved and the manuscript is now comprehensible. However, the manuscript is still not suitable for publication in its current form. Main problems I found are listed here:

Pasquet (Colloids and Surfaces A: Physicochemical and Engineering Aspects 2014) showed the effect of Zn2+ ions coming from dissolution of ZnO nanoparticles. Later Matula et al. (Soft Matter 2016) also showed this in respect to ZnO nanoparticles of various shape. There are two main aspects that were not considered within manuscript under review. 1) ZnO IS dissolving in water and water based media. 2) Gram-negative (G-) strains possess LPS layer, which provide protection against toxic agents and ions (see also Li, D. Lee, X. Sheng, R.E. Cohen, M.F. Rubner, Langmuir, 2006, 22, 9820-9823). Due to lack of LPS layer, Gram-positive (G+) strains are susceptible to presence of free zinc ions. Gram-positive strains are more susceptible to disturbances of ionic homeostasis.

In Figure 8 there are control experiments aiming to evaluate the effect of Zn2+ but in both cases (G+ and G-) there is no apparent effect. This is suspicious, especially in view of relatively large inhibition zones of ZnO NPs. These inhibition zones might be due to 1) reactive oxygen species (ROS), 2) diffusion of whole ZnO NP, 3) effect of eugenol, 4) release of zinc ions.

Ad 1) ROS are generated upon light illumination (which Authors did not mention in the text). It is very unlikely that there is some kind of bystander effect (death transmitted by the release of the cell content as an avalanche) as there is usually NO inhibition zone for immobilized antibacterial.

Ad 2) Blobs as large as presented in Figure 5 are practically immobile in agar gel.

Ad 3) I eliminate eugenol as ZnO NPs were calcinated in 400 C, what most likely resulted in decomposition of all organic compounds.

Ad 4) what left only Zn2+ and there is a clear discrepancy (in my opinion) between control and ZnO NP (also in view of published reports).

Such large nanoparticles WILL NOT get past cell envelope of the bacteria. Bacteria do not perform endocytosis and can uptake only very small particles (few nm). Figure 5 and Figure 6 present electron micrographs of studied ZnO NP. Again in my opinion these are aggregates. True nanoparticles should look something like in abovementioned paper by Matula et al. (Figure 1, Soft Matter 2016). I agree that XRD will give smaller values as it measured crystalline grains within aggregates. Authors synthetized ZnO NP (a) to (e) (differing in amount of C. verum used for the synthesis) but did not link some results to any of these (e.g. Figure 8). There are still mistakes in the text, e.g. P4L153 “(…) complex (…) undergoes reduction during oxidation” – I understand that something needs to be reduce for something else to be oxidized, but this sentence is not acceptable. There should be raw data provided as Supporting Info for MIC evaluation. I am still not sure if the values are correct (especially as in second Table 1 there are old values given (P11, 4 ug/ml and 31 ug/ml) Micro stress is presented in Table 1 but not mentioned in the text. There are two Table 1 in the main text. There are three Figure 8 in the main text. SEM picture does not support statement on P5L178-179. Some terms are not defined e.g. SDW, W-H etc. Why Figure S2 does not show eugenol? There is no Supplementary Table 2, to which Authors refer on P9. Actually there is not even Supplementary table 1 provided (and it is not mentioned in fact). Figure 1 and Figure 8 (third) are not needed. Introduction lacks of short summary on antibacterial effect of ZnO nanostructures on bacteria. Second sentence of the Introduction section is controversial. It is known for many years that decreasing the size of the portion of matter increases its cytotoxicity (see Net el al. 2011)

Author Response

We profusely thank the reviewers for their constructive comments. Herewith we are submitting the revised manuscript following incorporation of all the suggestions as indicated by the reviewers. We hope the reviewers will be happy with the corrections incorporated. In the revised manuscript, the questions raised by editor/ reviewers’ have been addressed and the changes are made in red color throughout the manuscript.

Comment No. 1: Pasquet (Colloids and Surfaces A: Physicochemical and Engineering Aspects 2014) showed the effect of Zn2+ ions coming from dissolution of ZnO nanoparticles. Later Matula et al. (Soft Matter 2016) also showed this in respect to ZnO nanoparticles of various shape. There are two main aspects that were not considered within manuscript under review. 1) ZnO IS dissolving in water and water based media. 2) Gram-negative (G-) strains possess LPS layer, which provide protection against toxic agents and ions (see also Li, D. Lee, X. Sheng, R.E. Cohen, M.F. Rubner, Langmuir, 2006, 22, 9820-9823). Due to lack of LPS layer, Gram-positive (G+) strains are susceptible to presence of free zinc ions. Gram-positive strains are more susceptible to disturbances of ionic homeostasis.

Answer: The authors would like to thank the reviewer for his keen observation and constructive comments. Here authors would like to state that the green synthesized ZnO-NPs were sonicated for 15 min before carrying out the antibacterial studies. After sonication, the aggregation of synthesized nanoparticles will be dispersed resulting in their separation and this might have had an inhibition effect during the interaction with the test pathogens. The inhibition offered by the nanoparticles might be due to the release of ROS, Zn2+ or by any other means which is still unknown. Here the authors would like to state that, the inhibition of pathogen growth occurred due to the generation of ROS upon interaction between the synthesized nanoparticles and the organism.

Comment No. 2: In Figure 8 there are control experiments aiming to evaluate the effect of Zn2+ but in both cases (G+ and G-) there is no apparent effect. This is suspicious, especially in view of relatively large inhibition zones of ZnO NPs. These inhibition zones might be due to 1) reactive oxygen species (ROS), 2) diffusion of whole ZnO NP, 3) effect of eugenol, 4) release of zinc ions.

Ad 1) ROS are generated upon light illumination (which Authors did not mention in the text). It is very unlikely that there is some kind of bystander effect (death transmitted by the release of the cell content as an avalanche) as there is usually NO inhibition zone for immobilized antibacterial.

Ad 2) Blobs as large as presented in Figure 5 are practically immobile in agar gel.

Ad 3) I eliminate eugenol as ZnO NPs were calcinated in 400 C, what most likely resulted in decomposition of all organic compounds.

Ad 4) what left only Zn2+ and there is a clear discrepancy (in my opinion) between control and ZnO NP (also in view of published reports).

Answer: As per the reviewer’s comments, we would like to state that the green synthesized ZnO-NPs were sonicated for 15 min before carrying out the antibacterial studies and hence there may be chances of all the above mentioned aspects (1 to 4) that may finally lead to their antibacterial effect.

Comment No. 3: Such large nanoparticles WILL NOT get past cell envelope of the bacteria. Bacteria do not perform endocytosis and can uptake only very small particles (few nm). Figure 5 and Figure 6 present electron micrographs of studied ZnO NP. Again in my opinion these are aggregates. True nanoparticles should look something like in above mentioned paper by Matula et al. (Figure 1, Soft Matter 2016). I agree that XRD will give smaller values as it measured crystalline grains within aggregates.

Answer: The authors would like to thank the reviewer for his keen observation and constructive comments and agree the same. Here authors would like to state that the green synthesized ZnO-NPs were sonicated for 15 min before carrying out the antibacterial studies while they were used as such for characterization except UV-vis analysis. The sonication of the particles is now included in the revised manuscript to clear all the confusions raised by the reviewer.

Comment No. 4: Authors synthesized ZnO NP (a) to (e) (differing in amount of C. verum used for the synthesis) but did not link some results to any of these (e.g. Figure 8).

Answer: As per the reviewer’s comments, we would like to mention here that ZnO-NPs (c) was selected based on the highest unit cell volume [V(Å)3].

Comment No. 5: There are still mistakes in the text, e.g. P4L153 “(…) complex (…) undergoes reduction during oxidation” – I understand that something needs to be reduce for something else to be oxidized, but this sentence is not acceptable.

Answer: As per the reviewer’s suggestion, the sentence in P4L153 has been modified according to the reviewer suggestion.

Comment No. 6: There should be raw data provided as Supporting Info for MIC evaluation. I am still not sure if the values are correct (especially as in second Table 1 there are old values given (P11, 4 ug/ml and 31 ug/ml)

Answer: The mistake in the values representation have been rectified in the revised manuscript. The photos of the MIC evaluation have been given as supplementary material (Suppl. Fig. 5) in the revised manuscript.

Comment No. 7: Micro stress is presented in Table 1 but not mentioned in the text. There are two Table 1 in the main text. There are three Figure 8 in the main text.

Answer: As per the reviewer’s suggestion, the microstress of the green synthesized ZnO-NPs have been presented in the text and all the mistakes in the quoting of figures and tables have been rectified in the revised manuscript accordingly.

Comment No. 8: SEM picture does not support statement on P5L178-179. Some terms are not defined e.g. SDW, W-H etc.

Answer: As per the reviewer’s suggestion, the statement in P5L178-179 has been deleted in the revised manuscript. The authors also like to state here that the terms (e.g. SDW, W-H, etc.,) have been given in full form wherever it has been quoted for the first time and thereon abbreviated in the revised manuscript.

Comment No. 9: Why Figure S2 does not show eugenol?

Answer: We would like to thank the reviewer for his keen observation and would like to state here that the GC-MS analysis has revealed the presence of eugenol (Figure S2). The authors would like to state that, the Figure S1 is the GC-MS chromatogram of the C. verum hexane extract wherein we peak values based on retention time (RT) and the peak at 11.49 RT corresponds to eugenol and the RT library of the peak 11.49 shows the presence of eugenol which can be seen in Figure S2.

Comment No. 10: There is no Supplementary Table 2, to which Authors refer on P9. Actually there is not even Supplementary table 1 provided (and it is not mentioned in fact).

Answer: All the mistakes in the quoting of figures and tables have been rectified in the revised manuscript accordingly.

Comment No. 11: Figure 1 and Figure 8 (third) are not needed.

Answer: As per the reviewer’s suggestion, Figure 1 and 8 have been removed in the revised manuscript.

Comment No. 12: Introduction lacks of short summary on antibacterial effect of ZnO nanostructures on bacteria. Second sentence of the Introduction section is controversial. It is known for many years that decreasing the size of the portion of matter increases its cytotoxicity (see Net el al. 2011)

Answer: As per the reviewer’s comments, a short summary on the antibacterial effect is added in the introduction part in the revised manuscript.

Reviewer 2 Report

the article has been improved, thanks!

however, I believe that, a chemist should see the article before getting into the final version.

new comments attached to the attached version, need to be considered carefully one by one

Author Response

We profusely thank the reviewers for their constructive comments. Herewith we are submitting the revised manuscript following incorporation of all the suggestions as indicated by the reviewers. We hope the reviewers will be happy with the corrections incorporated. In the revised manuscript, the questions raised by editor/ reviewers’ have been addressed and the changes are made in red color throughout the manuscript.

Comment No. 1: Re-check again this expression, enhanced of what???, if the hexane extract was not active!

Answer: The title has been modified according to reviewer’s suggestion in the revised manuscript.

Comment No. 2: which techniques used to see the trapped –OH?

Answer: The sentence has been removed and modified for the better understanding of the sentence in the revised manuscript as per the reviewer’s suggestion.

Comment No. 3: what is the function of the TEM, if HR-TEM employed?

Answer: Here the HR-TEM was employed to know the D-spacing of the green synthesized ZnO-NPs.

Comment No. 4: is far from reality, after heating at 400 degree, all organic compounds combusted, nothing left except the ZnO NPs 

Answer: We agree with the reviewer’s comment and the sentence has been modified accordingly in the revised manuscript.

Comment No. 5: this paragraph is plagiarized from ref 's 38, 39

Answer: The paragraph which was plagiarized with Ref.: 38 and 39 has been modified in the revised manuscript.

Comment No. 6: the analysis given is quite confusing, and we cannot relay on such data, no separation, peak tailing and no resolution, not sure if the conditions or the column is bad (or both of them).

Answer: We would like to state here that, the GC-MS analysis was carried out with standard procedure and we have provided the raw data as Fig. S1 and S2. We like to inform that, if needed we will provide all the raw data obtained during the GC-MS analysis as they have provided PDF files of the same.

Comment No. 7: this may need more elaboration (need experimental evidences), if the Zn (metal) formed, it doesn't mean the oxide is formed even under the atmospheric O2. however if this is correct, the authors need to explain why the long term heating at high degree takes place, and need to check the ZnO formation before heating.

Answer: The author’s agree with the reviewer’s comment. We would like to state here that, the contradictory statement mentioned in P4L153-154 (presently P4L166-167) has been removed and modified for better understanding of the same.

Comment No. 8: after calcination, no organic compound will remain (check the given IR, no signals for organic functional group there). also, be careful, the double bond NEVER make coordinate covalent bond as given in the figure (is wrong), please keep only the bond with the oxygen atom

Answer: We agree with the reviewer’s comments and the mistake in the Fig. 2 has been modified accordingly in the revised manuscript.

Comment No. 9: the data from SEM and/or TEM is sensitive and depend on the area of the given photo, which in most cases not representative to the whole NPs, if the authors discuss the concentration dependency they need to make sure other areas as well, not one place.

Answer: The author’s agree with the reviewer’s comment. We would like to state here that, the contradictory statement mentioned in P5L178-179 (presently P4L166-167) has been removed and modified for better understanding of the same.

Comment No. 10: there is no any band in the IR confirming what the authors speaking about????

Answer: As per the reviewer’s suggestion, the FT-IR spectra of the plant extract is provided as a supplementary figure (Suppl. Fig. 4A) in the revised manuscript wherein we can find broad spectra at 3300 to 3200 cm-1.

Comment No. 11: another photo required with HRTEM, high resolution (at say 10-20 nm). TEM is not clear

Answer: We regret to inform that, the time required for TEM analysis to provide a high resolution TEM image was not possible and we tried our level best to make the necessary arrangement as the instrument was not available/ working condition. We assure here that, if the journal provides at least 30 days time we will assure that the same will be provided if needed.

Comment No. 12: authors mentioned above the potential of eugenol as antibacterial agent, any explanation; why the total hexane extract not showing antibacterial activities???

Answer: From the literature survey it was noted that eugenol a secondary metabolite of C. verum is known to possess many biological activities including antibacterial potentiality. Further, eugenol in pure form possesses antibacterial activity. In our studies, the hexane extract of C. verum bark extract possessed a fraction of eugenol and this was taken only to depict an plausible mechanism which might have contributed to the antibacterial potentiality of the green synthesized ZnO-NPs. Here we would like to confirm that we are not exactly predicting that eugenol has lead to the antibacterial potentiality but might have contributed for the same.

Comment No. 13: which sample of the prepared NPs was selected

Answer: In the study, ZnO-NPs(c) {based on the highest unit cell volume [V(Å)3]} were used to evaluate its antibacterial efficacy and the same has been mentioned in the revised manuscript.

Comment No. 14: is it from the first draft?

Answer: The mistake from the first draft is rectified in the revised manuscript as per the reviewer’s suggestion.

This manuscript is a resubmission of an earlier submission. The following is a list of the peer review reports and author responses from that submission.

Round 1

Reviewer 1 Report

The manuscript entitled „Cinnamomum verum bark extract mediated green synthesis of ZnO nanoparticles and characterization of their enhanced antibacterial activity” authored by Lakshmeesha T.R, Mohammad Azam Ansari, Suriya Rehman, Daruka Prasad, Udayashankar A.C, M. Murali, Mohammad A. Alzohairy, Ahmad Almatroudi, Ashwini B.S, Amruthesh K.N, Chandra Nayak, and Niranjana SR is not suitable for publication and I cannot see any means of improving it to achieve standards of any indexed journal. The reasons for such harsh opinion are as follows:

- Some parts of the manuscript are incomprehensible. I did my best to understand what Authors really wanted to convey, but sometimes it was just impossible to even guess the true meaning.

- There are numerous things that are just wrong. For instance Figure 2 is wrong on so many levels. How come there are carbon atoms with two double bonds with neighboring carbons? The dashed lines seems to be completely random, connecting ZnO with sp2, sp3 carbon atoms or oxygen. Sometimes single carbon atom is connected with two ZnO domains (?). This is NOT how egg box model works. Also ZnO NP are MUCH larger than this mesh, so how eugenol can trap numerous NPs? I believe that this was attempt to show how organic layer is protecting ZnO core, but this part is completely misleading.

- From the presented data it seems that Authors did not obtain nanoparticles. The presented structures are polycrystalline aggregates of sizes in range of few um. Only PXRD gave the nano-size, as it shows the sizes of crystalline domains within larger entities. By no means one can call it nanoparticles. The band gap supporting sizes in range of nanometers was established from absorbance spectra, which are of very poor quality. This shape is not what one can expect for ZnO nanoparticles.

- Presented data showing antibacterial properties are misinterpreted. From Figure 7 is seems that as high concentration as 2 mg/ml is lower than MIC. Minimum inhibitory concentration (MIC) is the lowest concentration, which prevents visible growth of bacteria. So above MIC inhibition should be 100%. On Figure 7 there is no such concentration showed. I have no clue where 4 ug/ml and 31 ug/ml came from. Microscopic pictures are completely not convincing if not analyzed deeper. SEM is not suitable for such analysis – the protocol of sample prep influence heavily what is visible under the microscope.

- There is no appropriate controls in this part of the manuscript. I believe that the antibacterial effect comes from plant extract (what should be addressed as control experiment) and not from these big blobs of ZnO. Some effect comes also from dissolution of ZnO and effect of Zn2+ - this is supported by higher susceptibility of G+ over G- bacteria. G- have LPS layer, which acts as scavenger, protecting bacteria against e.g. ions.

There are also other, smaller issues, with the manuscript. It is just sloppy. Even the list of authors is wrong – some authors are given with first and last names, some with initials without coma and some with coma in between, some with initials before and some after last name. This is first time I see such level of zero effort to comply with submission rules.

Reviewer 2 Report

The authors need to be careful before re-submitting this work for publication:

1- first the hexane extract never contains polysaccharides and/or proteins.

2- the hexane is quite flammable petroleum fraction can't be left at 400 degree oven.

3- the model mentioned to explain the NPs formation has nothing to do with this case of study.

4- the GC-MS is not right

5- the design of the experiment from the first step is not perfect and use some expression not applicable like "stoichiometric" 

Reviewer 3 Report

This experimental work is well organized and presented. The novelty of this study work is clearly seen in the biosynthesis of ZnO NPs using C.verum extract. 

There is sufficiently detailed discussion and comparison with previously published data on the biosynthesis of ZnO NPs using other plant extracts and their antibacterial use.